# Patient Satisfaction and Impact on Oral Health after Maxillary Rehabilitation Using a Personalized Additively Manufactured Subperiosteal Jaw Implant (AMSJI)

**DOI:** 10.3390/jpm13020297

**Published:** 2023-02-08

**Authors:** Casper Van den Borre, Björn De Neef, Natalie A. J. Loomans, Marco Rinaldi, Erik Nout, Peter Bouvry, Ignace Naert, Maurice Y. Mommaerts

**Affiliations:** 1Doctoral School of Life Sciences and Medicine, Vrije Universiteit Brussel, 1090 Brussels, Belgium; 2European Face Centre, Universitair Ziekenhuis Brussel, Vrije Universiteit Brussel, 1090 Brussels, Belgium; 3Department of Oro-Maxillo-Facial Surgery, General Hospital Oudenaarde, 9700 Oudenaarde, Belgium; 4Private Clinic Face Ahead Antwerp, 2000 Antwerp, Belgium; 5Division of Oro-Maxillo-Facial Surgery, GZA Hospitals, 2000 Antwerp, Belgium; 6Private Practice, 40100 Bologna, Italy; 7Oral and Maxillofacial Surgery, ETZ Hospitals, 5022 GC Tilburg, The Netherlands; 8Department of Oro-Maxillo-Facial Surgery, AZ Alma, 9900 Eeklo, Belgium; 9Department of Prosthetic Dentistry, KU Leuven, 3000 Leuven, Belgium

**Keywords:** three-dimensional printing, subperiosteal, implant, patient satisfaction, alveolar bone loss, patient-specific implants

## Abstract

Subperiosteal implants (SIs) were first developed by Dahl in 1941 for oral rehabilitation in case of severe jaw atrophy. Over time, this technique was abandoned due to the high success rate of endosseous implants. The emergence of patient-specific implants and modern dentistry allowed a revisitation of this 80-year-old concept resulting in a novel “high-tech” SI implant. This study evaluates the clinical outcomes in forty patients after maxillary rehabilitation with an additively manufactured subperiosteal jaw implant (AMSJI^®^). The oral health impact profile-14 (OHIP-14) and numerical rating (NRS) scale were used to assess patient satisfaction and evaluate oral health. In total, fifteen men (mean age: 64.62 years, SD ± 6.75 years) and twenty-five women (mean age: 65.24 years, SD ± 6.77 years) were included, with a mean follow-up time of 917 days (SD ± 306.89 days) after AMSJI installation. Patients reported a mean OHIP-14 of 4.20 (SD ± 7.10) and a mean overall satisfaction based on the NRS of 52.25 (SD ± 4.00). Prosthetic rehabilitation was achieved in all patients. AMSJI is a valuable treatment option for patients with extreme jaw atrophy. Patients enjoy treatment benefits resulting in high patient satisfaction rates and impact on oral health.

## 1. Introduction

Masticatory rehabilitation of the severely atrophied maxilla has always been a difficult problem. Historically, preprosthetic surgical techniques, including absolute and relative augmentations, have been used to improve the retention of traditional removable dentures. Dahl developed the subperiosteal implant (SI) to support the denture and improve masticatory function [1]. However, these “classical” sub-periosteal implants did have a good reputation. The reasons for failure were plentiful. Vitalium^®^, a cobalt chrome alloy that is inert in human tissue, was used in the form of a frame; sometimes, the resectioning of keratinized mucosa was performed around the tissue piercing posts, and over-the-mucosa impression techniques were performed, leading to an inadequate fit of the SI [1,2]. As a result, poor osseointegration occurred in addition to soft tissue dehiscence, pathological pocket formation and infection, ultimately leading to the failure of the entire SI system, leaving considerable bone defects.

To improve the survival and success rates, numerous changes have been made to the technique and design of SIs over the years, affecting both subgingival and supragingival structures. The emergence of modern dentistry improved medical imaging and fitting, the 3D printing of titanium, and improved material knowledge enabled a revision of the 80-year-old concept of subperiosteal implants, resulting in a new ‘high-tech’ subperiosteal implant [3,4]: the additively manufactured Subperiosteal Jaw Implant (AMSJI) (see Figure 1). AMSJI is a patient-specific, custom 3D-printed implant for immediate functional recovery with just one procedure under local, sedation, or general anesthesia.

Patient-specific SIs are re-emerging and are regularly used at present in clinical practice [5,6,7]. Several long-term data have been published on the survival rates of traditional SI patients [8,9]. However, patients’ perspectives were often not considered and studies evaluating patient-related outcomes following SIs are rare. One study reported excellent results at 1 year in a small group of maxillary AMSJI patients with limited follow-up outcomes [10]. High patient satisfaction is an essential goal to be achieved in oral rehabilitation. By measuring patient-related outcomes, the true treatment benefit (patient satisfaction) can be evaluated and therefore cannot be ignored. The aim of this study is to collect data on patient-reported satisfaction and to score the impact on oral health in patients with AMSJI in the severely atrophic maxilla in a larger group of patients, treated by experienced surgeons, with a follow-up session in the medium term, and to compare it with the commonly used methods of oral rehabilitation at present.

## 2. Materials and Methods

An international multicenter study was set up and included a total of 40 patients, of which 31 patients were Belgian, 5 Italian, and 4 Dutch. Surgeons experienced in the technique with more than five patients treated with AMSJI were approached to participate in the study. The inclusion criteria were all patients who underwent bilateral maxillary AMSJI placements at least one year ago. In total, 122 patients were eligible for inclusion; however, the number was limited by patient and surgeon decisions to enroll in this retrospective study. All AMSJIs were placed for maxillary severe atrophy (Cawood-Howell classification 5 or higher). Maxillary defect reconstructions were excluded; no other exclusion criteria were used.

All patients were evaluated using a survey that was anonymized using a “patient code”. This was randomly chosen and not linked to the patient or hospital. Broad demographic information was obtained alongside subjective data on patient satisfaction and impact on oral health. Two questionnaires were used:A.The Oral Health Impact Profile-14 (OHIP-14)

The OHIP-14 includes seven domains related to functional limitations, physical pain, psychological discomfort, and physical, psychological, and social disabilities. Each domain consists of two questions scored on a five-point scale: 0, never; 1, almost never; 2, occasionally; 3, often; and 4, very often or every day. Domain scores were obtained by summing the answers to the two corresponding questions. Total scores were derived by summing all scores of all 14 questions. The score can range from 0 to 56 with domain scores ranging from 0 to 8. The higher the OHIP-14 score, the worse the oral health-related quality of life (OHRQoL).

B.Numerical Rating Scale (NRS)

The NRS is based on the visual analog scale (VAS) and aims to provide a greater insight into aesthetic benefit, chewing, comfort, phonetics, cleaning, and overall satisfaction. This scale consists of six questions with an eleven-point scale ranging from “0” for “not at all satisfied” to “10” for “very satisfied”. Adding the scores from all six questions results in a total score that can range from 0 to 60, where 0 is the worst and 60 is the highest possible satisfaction score.

### Statistical Analysis

The data were analyzed using SPSS version 26.0. (IBM, New York, NY, USA) for Mac OSMojave. The means and standard deviations were calculated for the OHIP-14 scores and NRS test. Each domain and question were also evaluated separately.

## 3. Results

Fifteen males (mean age: 64.62 years, SD ± 6.75 years) and twenty-five females (mean age: 65.24 years, SD ± 6.77 years) with a mean follow-up period of 917 days after AMSJI installation (SD ± 306.89 days) were included in this study. The final restoration of the prosthesis was successful in all patients and all patients presented with their fixed or removable prostheses in use at the time of consultation. There were 12 patients with postoperative inflammation (i.e., swelling, marked redness, pain, etc.). All were initially treated with antibiotics. Due to an apparent soft tissue infection, drainage, exploration and/or mechanical debridement was performed in six of these patients. In three patients, a post had to be removed due to persistent and uncontrollable infections (see Figure 2 and Figure 3). The stability of the AMSJI implant or prosthetic restoration was not compromised in these patients. At the time of examination, all but one of the AMSJI implants were firmly fixed (mobility of >1 mm after removal of the final restoration). Partial exposure of the arms was observed in 26 patients; however, patients did not experience this as a functional or aesthetic impediment.

Total OHIP-14 was calculated to provide an overall picture at the time of the interview. A mean value of 4.20 was calculated (SD ± 7.09). An evaluation of each domain was performed separately (see Table 1). Patients reported a mean NRS scale value of 52.25 (SD ± 4.00). Mean scores based on each domain/question separately were also calculated, presenting a more thorough representation (see Table 2). A graphic representation of the data set for OHIP-14 and the NRS scale is presented in Figure 4.

## 4. Discussion

Complete edentulism has been a major problem for a long period of time and is often described as the “final marker of disease burden for oral health” [11]. Although the prevalence of edentulism has decreased in recent decades, it is still considered a major problem worldwide [12]. One of the associated problems of edentulism is the significant effect on residual ridge resorption. The alveolar ridge of patients who remain edentulous for a long time becomes vestigial due to bone resorption [13]. This process is enhanced further by the adverse forces created when loading the jaws with soft tissue-supported dentures [14]. Continued resorption can result in ill-fitting dentures, leading to retention problems that compromise mastication and speech, and cause functional and sensory disturbances in the oral mucosa, salivary glands, and musculature [15].

Oral rehabilitation using endosseous implants has become a standard treatment option. However, due to severe resorption, the placement of endosseous implants is not always possible. Autologous bone augmentation techniques represent the “gold standard” for restoring alveolar ridge bone volume. One of the preferred donor sites, in case of reconstruction of large deficiencies (as is the case with a Cawood Howell class V or more), is the iliac crest. Gjerde et al. (2020) assessed patient-reported outcomes in 44 patients (mean age of 61.2 years ± 13) following maxillary alveolar ridge augmentation with anterior iliac crest grafting. An OHIP-14 score of 8.4 ± 9.7 has been reported [16]. The functional disability domain scored the highest (2.34) and the social disability domain scored the lowest (0.61). This is in accordance with our study. “Functional limitation” (1.08) and “Physical pain” (1.00) were indeed graded the highest. One of the main reasons was that a limited number of patients still had minor pronunciation problems. Non reported painful aching; however, some still needed some adaptation time to get used to their final prosthesis. Social disability (0.25) and handicap (0.23) scored the lowest as almost none of the patients reported any signs of being more irritable with other people because of their AMSJI installation. None of the patients reported any decrease in life satisfaction at the time of the investigation.

The calvarial bone serves as a valuable alternative to iliac crest bone. Wortmann et al. (2022) conducted a meta-analysis and compared patient-reported outcomes following autogenous iliac bone or calvarial bone harvesting in orally compromised patients [17]. They obtained patient-reported satisfaction with a median VAS score ranging from 8.8–10 in 206 patients following calvarial bone augmentation. For anterior iliac bone grafts, 696 patients were enrolled, and overall patient satisfaction was reported: the median VAS score ranged from 9.5 to 10. No statistical difference was observed when two techniques were compared.

Patient-related outcome measures for AMSJI are comparable to the average satisfaction rates of autogenic bone augmentation. However, AMSJI requires only one surgical procedure and provides immediate postoperative chewing function. This contrasts with bone regeneration techniques that use a two-step protocol. The first augmentation must occur and endosseous implants cannot be placed until three to four months later, that is, if the resorption of the graft has not occurred. Time is then required for the implants to integrate into the bone, further delaying the final placement of the prosthesis. Between stages, patients are advised not to wear dentures for a period of time in order not to compromise the graft and to ensure proper healing. Another drawback is that harvesting extraoral bone grafts for ridge augmentation is complex to perform and very technique-dependent. Gjerde et al. (2022) reported only 70.1% implant survival along with prosthetic rehabilitation after 1 year. Two patients (4.7%) reported that their oral health deteriorated after treatment. Three patients (7.30%) reported walking difficulties. Donor-site pain was reported by 16 patients (38%) and lasted on average for 18.10 ± 16.10 days. In addition, patients had an average of 4.3 days of hospitalization and 20.2 days of sick leave after iliac crest-derived alveolar bone grafting [16].

Another common option and alternative for the rehabilitation of the atrophic maxilla are zygomatic fixtures. Several studies indicated a high success rate and predictability [18,19,20,21]. However, there are no clinically applicable criteria for success and the definition of “success” is used as a very flexible term. Most studies consider success to be the survival of the implants placed. Objective reporting of patient satisfaction and quality of life over time is often absent or even ignored when reporting outcomes. An exception is the study by Fernández-Ruiz et al. (2021). These authors examined the quality of life and satisfaction in 40 patients who were rehabilitated with fixed prostheses supported by a combination of zygoma fixtures and conventional implants (anterior region) [19]. Patients’ follow-up treatments were 19.40 ± 4.37 months and a mean VAS of 18.48 ± 3.42 was reported. Although reasonably good patient satisfaction scores are reported, this article should be read with caution. A recent review of this article in the “Journal of Evidence-Based Dental Practice” found that this study had a high risk of bias, which minimizes the applicability of the results [22].

Compared to zygomatic fixtures and autogenous bone augmentation, AMSJI is a more patient-friendly alternative. Patients can be treated in an outpatient clinical setting with local anesthesia alone (for those who so desire). No hospitalization is required, and patients often report only mild pain that is easily controlled with first-line analgesics (acetaminophen and NSAIDs). Postoperative complications were observed. However, these cannot be compared to the major complications (i.e., penetration into the eye socket) that occur in some cases following the placement of the zygomatic implant.

One of the limitations of this study was that no baseline value, neither data before nor after SI installation, were available as this was a non-prospective study. For this reason, it is not possible to compare or calculate any statistical differences before or after AMSJI installation. However, the goal was to evaluate patients’ satisfaction and impact on oral health at the time of investigation, and to compare these to the techniques used at present. Few studies exist that calculate PROMS and OHIP-14 values in the general population. In a previous study by Dahl et al. (2011), an OHIP-14 score of 4.1 was observed in the general Norwegian adult population. Considering this as a representative value for the general population, the patients in our study reported almost equal OHRQoL values [23]. The same was observed by Wang et al. (2021) who evaluated patient satisfaction and quality of life related to oral health 10 years following the placement of endosseous implants in a non-atrophied alveolar ridge [24]. They observed that patients were almost as satisfied as the natural-teeth population in terms of function and aesthetics. The low OHIP-14 and high NRS scores in our series may be explained by the fact that the included patients were all orally crippled and had almost no alveolar ridge to maintain a prosthetic construct. Patients were bound by relining sessions and denture adhesives to improve stability during normal functioning. Any improvement in function would likely have a major positive impact and OHRQoL. Most AMSJI patients had some rehabilitation problems in the past with various failed augmentation techniques. It is therefore quite understandable that these patients were very satisfied to finally receive permanent teeth.

## 5. Conclusions

Oral rehabilitation in patients with severe maxillary atrophy using a personalized AMSJI is a valuable alternative to bone augmentation procedures and zygomatic/pterygoid implants. Although some complications were reported, patients enjoyed the treatment benefits resulting in high patient satisfaction and impact on oral health.

## Figures and Tables

**Figure 1 jpm-13-00297-f001:**
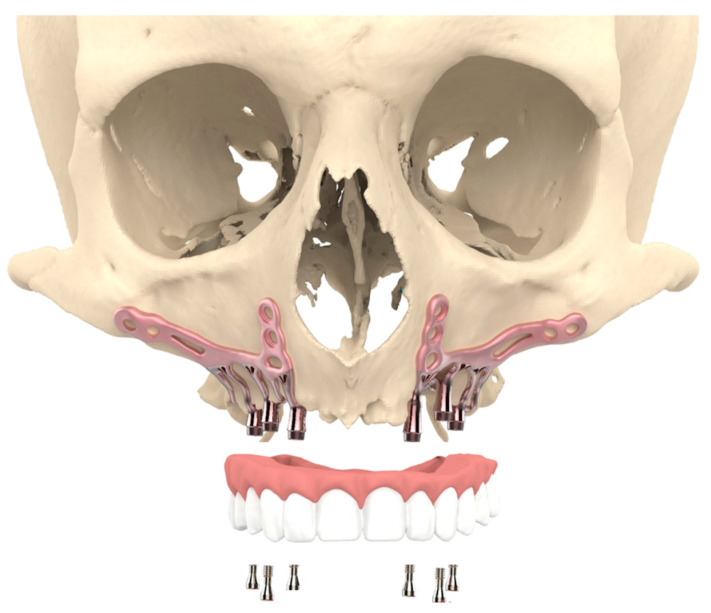
Visualization of the patient-specific AMSJI for the maxilla.

**Figure 2 jpm-13-00297-f002:**
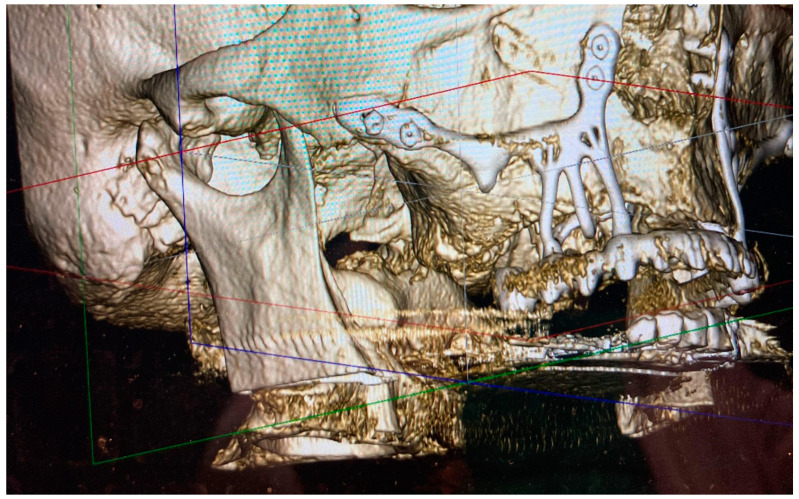
Cone-beam-computed tomography and three-dimensional reconstructed image of one of the patients where the distal arm of the right AMSJI had to be removed due to persisting infection.

**Figure 3 jpm-13-00297-f003:**
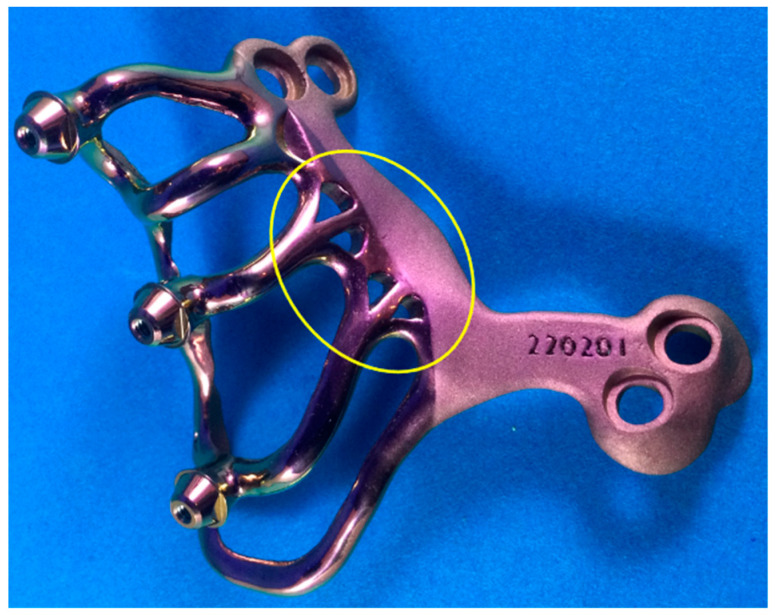
Picture of an AMSJI subunit with detailed visualization of the branched structure (yellow circle) connecting the basal looped frame with the arms and posts. In case of uncontrolled infection, a post can be easily removed by cutting these branches, without affecting the other arms.

**Figure 4 jpm-13-00297-f004:**
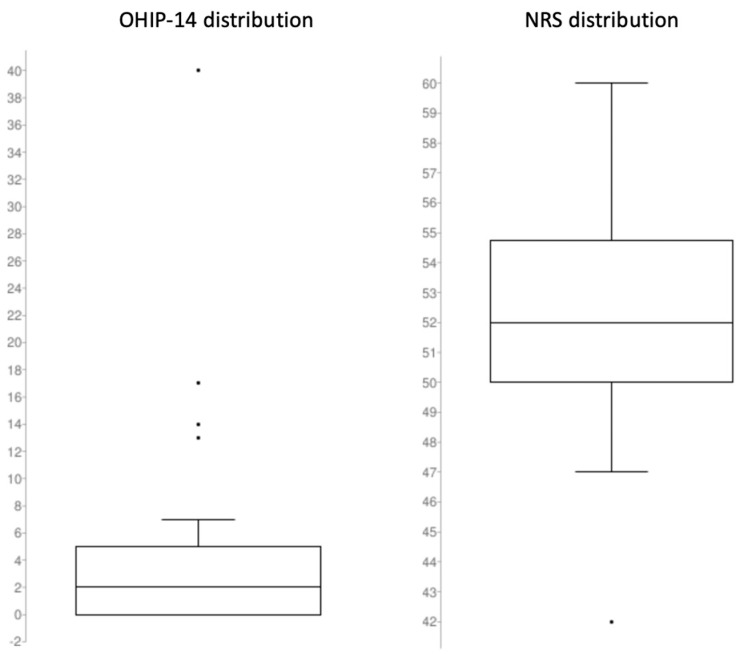
Visual representation of the distribution of the data set of the OHIP-14 and NRS score. (**Left**): The boxplot of the OHIP-14 values ranges from 0 (first quartile) to 5 (third quartile). Median value is 2 and interquartile range is 5. Minimal and maximal values were, respectively, 0 and 40, with 40, 17, 14, and 13 being the outliers. (**Right**): The boxplot of the NRS values ranges from 50 (first quartile) to 54.75 (third quartile). Median value is 52 and interquartile range is 4.75. Minimal and maximal values were, respectively, 42 and 60, with 42 being the only outlier.

**Table 1 jpm-13-00297-t001:** Results of the Oral Health Impact Profile-14 (OHIP-14).

Domain	Mean	SD
Overall OHIP-14	4.20	7.09
1. Functional limitation	1.08	1.51
2. Physical pain	1.00	1.75
3. Psychological discomfort	0.75	1.45
4. Physical discomfort	0.53	1.20
5. Psychological disability	0.38	1.13
6. Social disability	0.25	0.84
7. Handicap	0.23	0.73

SD, standard deviation; overall OHIP-14 is provided together with values of each domain separately. A low mean OHIP score of 4.20 (SD ± 7.09) was calculated, indicating a high oral health-related quality of life.

**Table 2 jpm-13-00297-t002:** Results of the Numerical Rating Scale (NRS).

Question	Mean	SD
Overall NRS	52.25	4.00
1. Aesthetic benefit	9.03	0.92
2. Chewing	8.83	1.11
3. Comfort	8.63	1.29
4. Phonetics	8.48	1.38
5. Cleaning	8.73	1.28
6. General satisfaction	8.58	1.11

SD, standard deviation; overall NRS is provided together with values of each question separately. A high mean NRS score of 52.25 (SD ± 4.00) is observed, indicating high patient satisfaction.

## Data Availability

The data presented in this study are available on request from the corresponding author.

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
