# Peer review of "Patient Satisfaction and Impact on Oral Health after Maxillary Rehabilitation Using a Personalized Additively Manufactured Subperiosteal Jaw Implant (AMSJI)"

_jpm, 2023, doi:10.3390/jpm13020297_

Round 1
Reviewer 1 Report
Additively manufactured subperiosteal jaw implant (AMSJI) is a valuable treatment option for patients with extreme jaw atrophy. Patients enjoy treatment benefits resulting in high patient satisfaction and impact on oral health. This cross-sectional study was well-written and simple with large sample numbers and a long follow-up time. There are several concerns regarding the manuscript.
1/ Please describe the distribution of OCHIP-14 and NRS data
2/ Please check the difference between male vs. female, young vs old, short vs long-time patients; small vs large defects... if applicable
3/ Discuss based on new analysis if applicable
Author Response
Dear Reviewer,
Please find the reply to your remarks in the attached PDF document

Reviewer 2 Report
The authors evaluated perceptions of satisfaction from AMSJI patients using OHIP-14 and NRS. The paper was straightforward and well-written. However, here are some minor suggestions for improvement.
1. The term "implant" is redundant after SI. (Line 46)
2. The first instance of the acronym PROMs should be spelled out. (Line 206)
3. The study has a lot of limitations. It would be best to enumerate this in the last paragraph. There could also be clinical recommendations that the authors may want to add.
4. The discussion part should also elaborate on the variables in Tables 1 and 2. For instance, "handicap" was less of a concern than "functional limitation".
Author Response
Dear Reviewer 2
The authors evaluated perceptions of satisfaction from AMSJI patients using OHIP-14 and NRS. The paper was straightforward and well-written. However, here are some minor suggestions for improvement.
- The term "implant" is redundant after SI. (Line 46)
Changed
- The first instance of the acronym PROMs should be spelled out. (Line 206)
Both OHIP-14 and OHRQoL are spelled out line 84 and 92.
- The study has a lot of limitations. It would be best to enumerate this in the last paragraph. There could also be clinical recommendations that the authors may want to add.
An extra paragraph is added at the end of the discussion
- The discussion part should also elaborate on the variables in Tables 1 and 2. For instance, "handicap" was less of a concern than "functional limitation".
The aim of this study was to evaluate patient-reported satisfaction and impact on oral health in patients rehabilitated with AMSJI in the severely atrophic maxilla and to compare it with current commonly used methods of oral rehabilitation. Several studies reporting on these alternatives, only use the OHIP-14 and NRS value in general. For these reasons we chose not to fully elaborate on the different variables. However, to accommodate the wishes of the reviewer we added a small paragraph line 225-231.
Reviewer 3 Report
This study describes an old problem with a very good solution approach. The first idea and description came from the working group around Rahlf et al and the idea was taken over here and examined with own patients. Since there is no solution for this type of patient, it is gratifying that other clinics are also dealing with the issue.
Author Response
Dear Reviewer 3
This study describes an old problem with a very good solution approach. The first idea and description came from the working group around Rahlf et al and the idea was taken over here and examined with own patients. Since there is no solution for this type of patient, it is gratifying that other clinics are also dealing with the issue.
Thank you very much ! Indeed, several working groups are studying the effect of the “novel” subperiosteal implants. We do want to point out that
- The first publication on AMSJI was
Mommaerts MY. Additively manufactured sub-periosteal jaw implants. Int J Oral Maxillofac Surg. 2017 Jul;46(7):938-940. doi: 10.1016/j.ijom.2017.02.002. Epub 2017 Mar 1. PMID: 28258795.
- The first publication of colleague Rahlf was
Gellrich NC, Rahlf B, Zimmerer R, Pott PC, Rana M. A new concept for implant-borne dental rehabilitation; how to overcome the biological weak-spot of conventional dental implants? Head Face Med. 2017 Sep 29;13(1):17. doi: 10.1186/s13005-017-0151-3. PMID: 28962664; PMCID: PMC5622522.
Note: July and September
Reviewer 4 Report
The researcher succeeded in evaluating the patient’s satisfaction and the effects on oral health after using a personalized additively manufactured subperiosteal jaw implant (AMSJI) for maxillary rehabilitation.
The article is relevant to the field of implantogy, because this theme has not been intensively analyzed until now.
The main purpose and objectives were clearly defined in the introduction of the article.
The visual representation of the statistics was very intuitively laid out.
The only downside worth mentioning is the lack of approval from the ethics committee.
All thigs considered, I would like to congratulate the whole team envolved in this research.
Author Response
Dear Reviewer 4
- The researcher succeeded in evaluating the patient’s satisfaction and the effects on oral health after using a personalized additively manufactured subperiosteal jaw implant (AMSJI) for maxillary rehabilitation. The article is relevant to the field of implantogy, because this theme has not been intensively analyzed until now. The main purpose and objectives were clearly defined in the introduction of the article. The visual representation of the statistics was very intuitively laid out. All thigs considered, I would like to congratulate the whole team envolved in this research.
Thank you very much
Round 2
Reviewer 1 Report
I recommend the manuscript for publication in its present form.
Thank you.